# Association between newborn separation, maternal consent and health outcomes: findings from a longitudinal survey in Kenya

Michelle Kao Nakphong [1,2] Emma Sacks,[3] James Opot,[4] May Sudhinaraset[1]

The Population Association of America Annual Meeting, May 5, 2021

¹Department of Community Health Sciences, University of California Los Angeles, Los Angeles, California, USA
²California Center for Population Research, University of California Los Angeles, Los Angeles, California, USA
³Department of International Health, Johns Hopkins University Bloomberg School of Public Health, Baltimore, Maryland, USA
⁴Innovations for Poverty Action, Nairobi, Kenya

**Correspondence to**
Michelle Kao Nakphong;
hmkao@ucla.edu

## ABSTRACT

**Objectives** Disrespectful and poor treatment of newborns such as unnecessary separation from parents or failure to obtain parental consent for medical procedures occurs at health facilities across contexts, but little research has investigated the prevalence, risk factors or associated outcomes. This study examined these experiences and associations with healthcare satisfaction, use and breast feeding.

**Design** Prospective cohort study.

**Setting** 3 public hospitals, 2 private hospitals, and 1 health centre/dispensary in Nairobi and Kiambu counties in Kenya.

**Participants** Data were collected from women who delivered in health facilities between September 2019 and January 2020. The sample included 1014 women surveyed at baseline and at least one follow-up at 2–4 or 10 weeks post partum.

**Primary and secondary outcome measures** (1) Outcomes related to satisfaction with care and care utilisation; (2) continuation of post-discharge newborn care practices such as breast feeding.

**Results** 17.6% of women reported newborn separation at the facility, of whom 71.9% were separated over 10 min. 44.9% felt separation was unnecessary and 8.4% reported not knowing the reason for separation. 59.9% reported consent was not obtained for procedures on their newborn. Women separated from their newborn (>10 min) were 44% less likely to be exclusively breast feeding at 2–4 weeks (adjusted OR (aOR)=0.56, 95% CI: 0.40 to 0.76). Obtaining consent for newborn procedures corresponded with 2.7 times greater likelihood of satisfaction with care (aOR=2.71, 95% CI: 1.67 to 4.41), 27% greater likelihood of postpartum visit attendance for self or newborn (aOR=1.27, 95% CI: 1.05 to 1.55), and 33% greater likelihood of exclusive breast feeding at 10 weeks (aOR=1.33, 95% CI: 1.10 to 1.62).

**Conclusions** Newborns, mothers and families have a right to high-quality, respectful care, including the ability to stay together, be informed and properly consent for care. The implications of these experiences on health outcomes a month or more after discharge illustrate the importance of a positive experience of postnatal care.

## Strengths and limitations of this study

⇒ This is one of the first studies to include survey questions surrounding details of separation of newborns from mothers and maternal consent for care, including risk factors and reasons for separation.

⇒ This study used longitudinal data collected over 10 weeks following delivery to assess associations with outcomes, including postpartum visit attendance and breast feeding.

⇒ Differences in participants who responded to the follow-up interviews 2–4 weeks and 10 weeks may have introduced bias due to differences in composition, thus comparisons of associations with outcomes between these two points should be interpreted conservatively.

⇒ Findings may have limited generalisability to other contexts, as the study sample included women who gave birth at facilities in Nairobi and Kiambu counties.

2.4 million children die within the first 28 days of life at a rate of 18 deaths per 1000 live births.[1] Kenya has made notable progress in reducing neonatal mortality in recent years,[2] but with its current neonatal mortality ratio of 19.6 deaths per 1000 live births, Kenya is unlikely to reach the 2030 Sustainable Development Goal target of 12 deaths per 1000 live births.[3] Yet, substantial progress can still be made: over 80% of newborn deaths are considered preventable, highlighting the critical need to improve the quality of maternal and newborn care.[4]

Efforts to enhance the quality of maternity care have drawn attention to women's experience of care, including respect and dignity. The WHO Vision on Quality of Care for maternal and newborn health outlines two essential, interlinked dimensions of quality of care: provision of care and experience of care.[5] Integrally, quality care, in addition to being safe, effective, timely and efficient, must also be equitable and people-centred. Experience

## INTRODUCTION

The first month of life constitutes the most vulnerable time for a child's survival. Globally,

of care includes effective communication, respectful and dignified treatment, and access to emotional support. Evidence indicates that across contexts, many women experience mistreatment and abuse,[6–11] and lack access to the support person(s) of their choice during labour and delivery.[12–14] Negative experiences of care not only contribute to dissatisfaction with care,[15–17] but may deter women from seeking future health services.[18–22] However, the corresponding elements of quality newborn care and the health impact of these experiences have been under-researched.

WHO recommendations for newborn care include skin-to-skin care, early and exclusive breast feeding, thermal and umbilical care, delayed bathing and maintaining the newborn with family as much as possible.[23] These practices are especially important for preterm and low birthweight infants, who are more vulnerable to mortality and morbidity. It is also recommended that families receive breastfeeding support, counselling on danger signs, and examinations of the infant and cord before discharge. Communication, consent and respect for autonomy are important components of a positive experience of care. But, efforts to improve the quality of newborn care have often focused on provision of care with low or no priority on experience of care.[24] Research on experiences of newborn care (parental consent for procedures or examinations, mistreatment and neglect, and separating infants from mothers or caregivers when not medically necessary) represents a notable gap in literature.

Evidence indicates that despite newborns' inability to exercise autonomy or use words to express their needs, positive experiences of care centre around their relationship and connection with their mothers and caregivers. A strong mother–infant relationship forms the foundation for infant health and development, and the period after birth is especially important.[25] Immediately following birth, infants display highly specialised attachment and bonding behaviour towards their mothers,[26 27] and sensory connections with mothers form a strong basis for the regulation of their physiological systems.[28] For example, research on skin-to-skin contact (SSC) after birth has shown that early sensory connections improve many physical, psychological and care outcomes, such as maintenance of newborn temperature,[29] stable respiratory and cardiac function,[30] organisation of sleep cycles,[31] development of neurophysiological systems[32 33] and reducing newborn stress.[29 34] In contrast, separation of mothers and infants hinders attachment, and induces stress and behaviours that alter physiological processes resulting in an altered developmental trajectory.[28 35]

This highlights newborns' needs to effectively bond with family, receive attentive care, and be kept warm and treated gently. Sensory cues from infants–suckling, sight, smell, sounds–also stimulate maternal neurobiological processes which subsequently affect maternal mental health and caregiving.[36–38] Evidence also suggests that the benefits of early bonding likely extend to other caregivers as well, such as fathers, with implications for future involvement in care.[39]

Fostering the mother–infant bond after birth also has profound implications for early and long-term breast feeding. Mother–infant interactions stimulate maternal milk production and prepare infants to breast feed by downregulating the neuroendocrine stress response and activating blood glucose regulation.[40 41] The evidence of a link between close maternal–infant contact and breast feeding is compelling: mother–infant pairs who engaged in SSC after birth were more likely to adequately suckle during their first feed, continue breast feeding after 1–4 months post-birth, score higher on suckling competence, and breast feed exclusively and for longer duration when compared with those without SSC.[29 42] This underscores the need to foster mother–infant interactions and minimise unnecessary interventions in maternity care.

Lack of information and informed consent has been cited as a factor contributing to traumatic experiences and dissatisfaction in maternity care.[43–45] Lack of consent for women's care is regarded as an essential component of respectful care, but consent for procedures performed on newborns has rarely been studied. Because infants are completely dependent on their caregivers, their caregivers are essential advocates to ensure safe and gentle medical care. Parents (or designated legal guardians) should be the decision-makers regarding their infants' care, and it is likely that women and families value the ability to make informed decisions and advocate for their newborns' care.

There are indications that poor treatment of newborns in facility-based maternity care occurs, including preliminary documentation of prevalence of certain types of occurrences. Recent literature has brought attention to the prevalence of practices such as unnecessary separation of newborns from mothers, neglect (absence of care) and non-consent for procedures.[7 10 46] However, these experiences have seldom been examined as primary indicators and available information has emerged from secondary data from other studies. A multicountry African study on treatment of women included observations of newborns in the first 2 hours after birth and found that over half of newborns were separated from their mothers within the first 2 hours, with higher prevalence among single and less educated women.[47] However, while some studies have documented high prevalence of these experiences and inequities in care, it remains unclear how they are associated with other measures of quality care or outcomes (eg, positive experiences of care, health behaviours or health outcomes). Moreover, evidence suggests that person-centred maternity care is consequential for newborn health. One recent study in Kenya found that women's poor experiences of care were associated with increased report of newborn complications and lower likelihood of attending future medical visits.[48] To our knowledge, no previous studies have linked disrespectful neonatal care experiences, namely separation of newborns from mothers and lack of parental consent for

newborn procedures, to health outcomes. This study aims to address these gaps.

Further research is needed to explore how newborn care experiences, and the implementation of institutional policies and provider practices, impact maternal and neonatal outcomes.[40 49] The aim of this study was to investigate the prevalence, risk factors and outcomes associated with newborn care experiences, namely separating newborns from mothers and parental consent for newborn care, in several facilities in Kenya. In this study, we examined these practices using data from the Strengthening Person-Centered, Accessibility, Respectful Care, and Quality (SPARQ) Study in Kenya. In addition to investigating the extent to which these experiences occur, we sought to understand underlying risk factors and whether certain groups were at greater risk (ie, 'discriminatory' practices). We also examined details surrounding newborn separation, including the duration and reasons for separation, and other newborn care practices such as SSC and breastfeeding support. Finally, we sought to understand the consequences of these experiences on maternal and newborn health outcomes: satisfaction with care, postpartum care attendance and breast feeding.

## METHODS

### Setting

This longitudinal study was conducted between September 2019 and January 2020 across six health facilities within Nairobi and Kiambu counties, Kenya. The facilities varied in type (three public hospitals, two private hospitals and one health centre/dispensary) and size (medium to large referral hospitals ranging from 100 to 900 reported deliveries per month). The facilities were conveniently selected based on their location within the two counties, reported deliveries of at least 100 per month and the facility administration's willingness to participate in the study.

### Data collection and recruitment

A team of 24 enumerators were involved in data collection, who had previous quantitative and research ethics training. The baseline survey was pretested with women who had recently delivered in four facilities (two of which were study facilities) to identify any issues with question flow or understanding via interviewing. The follow-up phone surveys were also pretested with this sample.

All surveys were administered in a private space within the facility to ensure confidentiality and privacy. Follow-up surveys were conducted by phone, ensuring privacy to maintain confidentiality.

Women were enrolled in the study post partum when recovering in the maternity ward or at discharge before exiting the facility, all within 48 hours after birth. Women provided written informed consent. Participants were also administered follow-up phone surveys between 2–4 weeks and/or 10 weeks post partum. Participants were given airtime credit of approximately $2.00 for the baseline and $0.50 for each follow-up as a token of appreciation.

### Analysis

The analytical sample included 1014 women who completed baseline and at least one follow-up interview (online supplemental file 1). Data for exclusive breast feeding were missing from three women in the follow-up 2–4 weeks and four women in the 10-week follow-up resulting in an analytical sample for exclusive breast feeding of 829 and 839 participants in the follow-up at 2–4 weeks and 10 weeks, respectively.

Variables examined are presented in table 1. Our outcomes of interest included *satisfaction with newborn care* measured at baseline, and *attendance of postpartum check-up* and *exclusive breast feeding* measured at follow-up. Primary independent variables were *newborn separation, newborn separation >10 min* and *consent for newborn procedures*. Although we did not find evidence in the literature about a critical duration of separation, we used a 10-minute threshold for separation as a proxy for short and long separation. Aside from *perceived necessary separation*, additional probing questions were not asked. We examined descriptive characteristics of all women in the sample. We explored sociodemographic characteristics of the participants including age, parity, marital status, education, employment, woman's birthplace, insurance status and facility type. We examined maternal health indicators of self-rated health status, maternal complications at delivery and report of maternal complications since discharge. Newborn health characteristics were also explored, including birth weight, gestational age, and report of newborn complications at baseline and after discharge. Lastly, we examined other measures of care received using a clinical quality index and a breastfeeding care index based on WHO standards of care.[23]

Bivariate analyses investigated relationships between newborn care indicators and outcomes using $X^2$ and t-tests. We conducted multivariate logistic regression analyses assessing associations between newborn separation and consent indicators and described outcomes. Analyses adjusted for factors that may theoretically confound associations including age, parity, marital status, education, employment, birthplace, health facility type, insurance status, self-rated health and clinical care index score. Analyses for *satisfaction with newborn care* also adjusted for maternal and newborn complications at baseline and analyses for outcomes measured at follow-up interview adjusted for maternal and newborn complications since discharge. Models for breastfeeding outcomes also adjusted for breastfeeding care index score. Cluster robust standard errors were used to account for clustering by facility. Model specification was tested using link tests and model fit was tested by Hosmer-Lemeshow goodness-of-fit tests. Sensitivity analyses examined potential effects of preterm birth and low birth weight, but no evidence of confounding was found. Analyses were performed using STATA/SE V.15.1.

**Table 1** Definition of variables

| Indicator | Question/measure | Values |
|---|---|---|
| *Outcome Indicators* | | |
| Satisfaction with newborn care | In general, how satisfied were you with the services and care your newborn received after delivery? (collected at baseline) | 'Satisfied or very satisfied' vs 'Dissatisfied or very dissatisfied' |
| Attendance of postpartum check-up | Have you attended any postpartum visits since being discharged?* (collected at follow-up) | 'Yes' vs 'No' |
| Exclusive breast feeding | How were you feeding your baby?* (collected at follow-up) | Fed infant with breastmilk only vs fed with any other substances |
| *Newborn separation and consent indicators* (collected at baseline) | | |
| Newborn separation | Was your baby ever separated from you by doctors, nurses or other health providers for any reason? | 'Yes' vs 'No' |
| Newborn separation >10 min | Was your baby ever separated from you for more than 10 min? | 'Yes' vs 'No' |
| Perceived necessary separation | Did you ever feel that it was not necessary for your baby to be separated from you? | 'Yes' vs 'No' |
| Consent sought for newborn procedures | Did the doctors, nurses or other providers ask you for your permission before doing procedures or examinations on your baby? | 'Yes' vs 'No' |
| *Other variables* (collected at baseline) | | |
| Clinical quality index | Summative score comprising 7 newborn care questions indicating whether specific procedures were performed (skin-to-skin contact after birth; infant examination after delivery; infant dried after birth; delayed bathing; cord examination; temperature assessment; and mother/family counselled on newborn danger signs). | Range 0–7 |
| Breastfeeding care index | Summative score of 3 indicators of breastfeeding support at the facility (ie, provider checked breast feeding within 2 hours of delivery; mother/family counselled about breast feeding; breast feeding observed or shown). | Range 0–3 |

*Postpartum visit could have been for either maternal or newborn health.

### Patient and public involvement

Outside of participating in pretesting of tools, patients and the general community were not involved in the design, recruitment or conduct of this study. During the consent process, participants were informed that they would not directly benefit from their participation or be involved in the dissemination of the study results. Study results will be shared with facilities in order to improve their service delivery.

### RESULTS

Descriptive characteristics of the sample of 1014 women, stratified by reports of newborn separation, newborn separation >10 min and consent sought for newborn procedures, are presented in table 2.

Women who were separated from their newborn more than 10 min tended to be slightly older (p=0.036), were of higher parity (p=0.021), and not born in Nairobi or Kiambu counties (p=0.043) compared with women reporting separation for less than 10 min or no separation. A greater proportion of women at government hospitals were separated from their newborns compared with women at government health clinics or private facilities (p<0.001). Delivery complications were also positively associated with newborn separation (p=0.004) and separation for more than 10 min (p=0.025).

Women who were asked permission for newborn procedures and examinations were older (p=0.026) and more likely to be covered under a health scheme or health insurance (p=0.028) than those who were not asked for permission.

Table 3 displays newborn characteristics and other clinical and breastfeeding care indicators by reported experience of newborn separation and consent sought for newborn care. A total of 17.6% of women reported being separated from their newborns while at the health facility. Among those women (n=178), the majority reported separation longer than 10 min (71.9%) and nearly half felt separation was unnecessary (44.9%). Most of those separated reported that their newborn was separated for procedures or examination (82.6%), but 8.4% reported that they did not know or were not told (Of those who reported 'Don't know', 80.0% reported being satisfied with newborn care. In comparison, 94.1% of the total

**Table 2** Descriptive characteristics of women included in the sample by reports of newborn separation, newborn separation for more than 10 min and consent sought for newborn procedures (n=1014)

| Characteristics | Total | Newborn separated | | | Newborn separated >10 min | | | Consent for newborn procedures* | | |
|---|---|---|---|---|---|---|---|---|---|---|
| | | Not separated | Separated | P value | Not separated/separated ≤10 min | Separated >10 min | P value | Did not ask for permission | Asked for permission | P value |
| Total number in group | 1014 | 836 | 178 | | 886 | 128 | | 607 | 406 | |
| Age (mean) | 25.7 | 25.58 | 26.11 | 0.202 | 25.55 | 26.54 | 0.036 | 25.38 | 26.1 | 0.026 |
| SD | (5.01) | (4.97) | (5.19) | | (4.95) | (5.32) | | (4.96) | (5.05) | |
| Parity (mean) | 2.0 | 2.00 | 2.16 | 0.050 | 2.01 | 2.22 | 0.021 | 2.02 | 2.04 | 0.714 |
| SD | (0.98) | (0.97) | (1.02) | | (0.97) | (1.03) | | (0.98) | (0.98) | |
| Multiparous | | | | | | | | | | |
| No | 35.7% | 36.7% | 30.9% | 0.141 | 36.7% | 28.9% | 0.086 | 36.1% | 35.2% | 0.780 |
| Yes | 64.3% | 63.3% | 69.1% | | 63.3% | 71.1% | | 63.9% | 64.8% | |
| Currently married or partnered | | | | | | | | | | |
| No | 81.8% | 18.7% | 16.3% | 0.458 | 18.2% | 18.8% | 0.874 | 19.1% | 17.0% | 0.393 |
| Yes | 18.2% | 81.3% | 83.7% | | 81.8% | 81.2% | | 80.9% | 83.0% | |
| Educational attainment | | | | | | | | | | |
| Primary or less | 43.7% | 43.8% | 43.3% | 0.894 | 44.0% | 41.4% | 0.855 | 44.8% | 41.9% | 0.267 |
| Vocational/secondary | 39.6% | 39.4% | 41.0% | | 39.4% | 41.4% | | 40.0% | 39.2% | |
| College/university | 16.7% | 16.9% | 15.7% | | 16.6% | 17.2% | | 15.2% | 19.0% | |
| Currently employed | | | | | | | | | | |
| No | 59.5% | 59.0% | 61.8% | 0.486 | 59.7% | 57.8% | 0.683 | 61.6% | 56.4% | 0.098 |
| Yes | 40.5% | 41.0% | 38.2% | | 40.3% | 42.2% | | 38.4% | 43.6% | |
| Born in Nairobi or Kiambu counties | | | | | | | | | | |
| No | 78.0% | 76.8% | 83.7% | 0.043 | 77.3% | 82.8% | 0.160 | 77.6% | 78.6% | 0.713 |
| Yes | 22.0% | 23.2% | 16.3% | | 22.7% | 17.2% | | 22.4% | 21.4% | |
| Self-rated health status | | | | | | | | | | |
| Excellent or very good | 34.7% | 33.5% | 40.4% | 0.043 | 34.1% | 39.1% | 0.144 | 33.3% | 36.9% | 0.502 |
| Good | 40.4% | 42.5% | 30.9% | | 41.6% | 32.0% | | 40.4% | 40.4% | |
| Fair | 15.7% | 15.2% | 18.0% | | 15.0% | 20.3% | | 16.8% | 14.0% | |
| Poor or very poor | 9.2% | 8.9% | 10.7% | | 9.3% | 8.6% | | 9.6% | 8.6% | |
| Maternal complications during delivery | | | | | | | | | | |
| No | 93.6% | 94.6% | 88.8% | 0.004 | 94.2% | 89.1% | 0.025 | 92.9% | 94.6% | 0.289 |

Continued

**Table 2** Continued

| Characteristics | Total | Newborn separated | | | Newborn separated >10 min | | | Consent for newborn procedures* | | |
|---|---|---|---|---|---|---|---|---|---|---|
| | | Not separated | Separated | P value | Not separated/ separated ≤10 min | Separated >10 min | P value | Did not ask for permission | Asked for permission | P value |
| Yes | 6.4% | 5.4% | 11.2% | | 5.8% | 10.9% | | 7.1% | 5.4% | |
| Reported maternal complications at follow-up | | | | | | | | | | |
| No | 86.1% | 86.4% | 84.8% | 0.592 | 86.0% | 86.7% | 0.827 | 85.3% | 87.2% | 0.403 |
| Yes | 13.9% | 13.6% | 15.2% | | 14.0% | 13.3% | | 14.7% | 12.8% | |
| Covered under health scheme or health insurance | | | | | | | | | | |
| No | 14.0% | 13.9% | 14.6% | 0.799 | 13.9% | 14.8% | 0.77 | 16.0% | 11.1% | 0.028 |
| Yes | 86.0% | 86.1% | 85.4% | | 86.1% | 85.2% | | 84.0% | 88.9% | |
| Facility type | | | | | | | | | | |
| Gov't hospital | 73.9% | 70.8% | 88.2% | <0.001 | 71.7% | 89.1% | <0.001 | 74.3% | 73.2% | 0.881 |
| Gov't health centre/ dispensary | 12.0% | 13.9% | 3.4% | | 13.4% | 2.3% | | 12.0% | 12.1% | |
| Private facility | 14.1% | 15.3% | 8.4% | | 14.9% | 8.6% | | 13.7% | 14.8% | |

*N=1013 (one refused to answer).

sample reported being satisfied with newborn care.). During separation, only one in five (19.1%) were able to have another parent, family member or caregiver remain present with the newborn. Almost two-thirds of women (59.9%) reported that health providers did not ask permission before performing procedures or examinations on their newborn.

Mothers of newborns with complications were more likely to report separation than those who reported no complications (p<0.001). Mothers who were separated from their newborns were more likely to have their infant examined after delivery than not (p=0.049) but were less likely to have a provider: ask consent before newborn procedures or examinations (p=0.023), check if breast feeding was going well (p=0.048), or observe or demonstrate how to breast feed (p=0.013). Almost all indicators of better clinical and breastfeeding care, such as SSC, newborn examination after delivery, newborn wiped, cord examination, temperature assessment, counselling on danger signs for newborns, and breastfeeding checks, counselling, and observation, were positively associated with consent sought for newborn procedures (online supplemental file 2). Among the full sample, the mean clinical quality index score was 4.14 (SD 1.37) out of 7. Those who reported being asked permission for newborn procedures had a mean clinical quality of care index of 4.64 (SD 1.17) compared with 3.81 (SD 1.38) among those who did not report being asked permission (p<0.001). The mean breastfeeding care index score among those who reported being asked permission was 2.3 (SD 0.94) out of 3 compared with 1.79 (SD 1.10) for those who reported not being asked permission (p<0.001).

The distributions of outcomes by reports of newborn separation, separation for more than 10 min and consent sought for newborn procedures are presented in table 4. Women who were asked permission for newborn procedures were more likely to be satisfied with care (p<0.001) and attend a postpartum visit after discharge (p<0.001) than those who were not asked permission.

Results of multivariate analyses examining associations between reports of newborn separation, consent sought for newborn procedures and health outcomes are presented in table 5 (full models are presented in online supplemental file 3). Among women reporting any newborn separation during care, there was a negative to no association with exclusive breast feeding at 2–4 weeks (adjusted OR (aOR)=0.67, 95% CI: 0.41 to 1.09). However, women who reported separation from their newborn longer than 10 min were 44% less likely to be exclusively breast feeding at 2–4 weeks compared with those reporting less than 10 min or no separation (aOR=0.56, 95% CI: 0.40 to 0.76). No associations were found with satisfaction with newborn care, postpartum visit attendance or exclusive breast feeding at 10 weeks.

Reported consent sought for newborn procedures was positively associated with satisfaction with newborn care at baseline, attending a postpartum visit by 10 weeks and exclusive breast feeding at 10 weeks. Women

**Table 3** Distributions of newborn characteristics and newborn care indicators by reports of newborn separation and newborn consent indicators (n=1014)

| Variable | Total | Newborn separation | | | Newborn separation >10min | | | Consent sought for newborn procedures* | | |
|---|---|---|---|---|---|---|---|---|---|---|
| | | Not separated | Separated | P value | Not separated or separated ≤10min | Separated >10min | P value | Did not ask for permission | Asked for permission | P value |
| Total number in group | 1014 | 836 | 178 | | 886 | 128 | | 607 | 406 | |
| **Health indicators** | | | | | | | | | | |
| Birth weight (BW) | | | | | | | | | | |
| Low BW (<2.5kg) | 2.3% | 2.2% | 2.8% | 0.002 | 2.3% | 2.3% | <0.001 | 2.8% | 1.5% | 0.588 |
| Normal BW (2.5–4.0kg) | 89.3% | 90.8% | 82.6% | | 90.6% | 80.5% | | 88.8% | 90.2% | |
| Macrosomia (>4.0kg) | 2.2% | 2.2% | 2.3% | | 2.1% | 2.3% | | 1.7% | 3.0% | |
| Missing | 6.2% | 4.9% | 12.4% | | 5.0% | 14.8% | | 6.8% | 5.4% | |
| Preterm | | | | | | | | | | |
| Not preterm (GA 38 weeks or more) | 77.4% | 77.2% | 78.7% | 0.418 | 77.3% | 78.1% | 0.266 | 77.9% | 76.9% | <0.001 |
| Yes, preterm (GA 37 weeks or less) | 22.4% | 22.7% | 20.8% | | 22.6% | 21.1% | | 21.9% | 23.2% | |
| Missing | 0.2% | 0.1% | 0.6% | | 0.1% | 0.8% | | 0.2% | 0.0% | |
| Newborn complications reported at baseline | | | | | | | | | | |
| Healthy, no issues | 96.3% | 97.5% | 90.5% | <0.001 | 97.5% | 87.5% | <0.001 | 96.4% | 96.1% | 0.929 |
| Complications reported | 3.6% | 2.4% | 9.6% | | 2.4% | 12.5% | | 3.5% | 3.9% | |
| Missing† | 0.1% | 0.1% | 0.0% | | 0.1% | 0.0% | | 0.2% | 0.0% | |
| Newborn complications reported since discharge | | | | | | | | | | |
| No complications | 67.8% | 68.2% | 65.7% | 0.525 | 68.1% | 65.6% | 0.582 | 68.0% | 67.2% | 0.790 |
| Complications | 32.3% | 31.8% | 34.3% | | 31.9% | 34.4% | | 32.0% | 32.8% | |
| **Newborn care indicators** | | | | | | | | | | |
| Newborn separation | | | | | | | | | | |
| No | 82.4% | – | – | | – | – | | 80.9% | 85.% | 0.093 |
| Yes | 17.6% | – | – | | – | – | | 19.1% | 15.% | |
| Consent sought for newborn procedures | | | | | | | | | | |
| No | 59.9% | 58.7% | 65.2% | 0.023 | 58.9% | 66.4% | 0.007 | – | – | |
| Yes | 40.0% | 41.3% | 34.3% | | 41.1% | 32.8% | | – | – | |
| Refused to answer | 0.1% | 0.0% | 0.6% | | 0.0% | 0.8% | | – | – | |
| **Of those who were separated from newborns (n=178)** | | | | | | | | | | |
| Newborn separation >10min | | | | | | | | | | |
| No | 4.9% | – | 28.1% | | – | – | | 26.7% | 31.1% | 0.534 |

Continued

**Table 3** Continued

| Variable | Total | Newborn separation | | | Newborn separation >10 min | | | Consent sought for newborn procedures* | | |
|---|---|---|---|---|---|---|---|---|---|---|
| | | Not separated | Separated | P value | Not separated or separated ≤10 min | Separated >10 min | P value | Did not ask for permission | Asked for permission | P value |
| Yes | 12.6% | – | 71.9% | | – | – | | 73.3% | 68.9% | |
| Was your baby separated from you by doctors, nurses or other health providers to perform procedures and examinations? | | | | | | | | | | |
| No | 1.6% | – | 9.0% | | 14.0% | 7.0% | 0.344 | 8.6% | 9.8% | 0.868 |
| Yes | 14.5% | – | 82.6% | | 78.0% | 84.4% | | 82.8% | 83.6% | |
| Don't know | 1.5% | – | 8.4% | | 8.0% | 8.6% | | 8.6% | 6.6% | |
| Did you ever feel that it was not necessary for your baby to be separated from you? | | | | | | | | | | |
| No | 9.7% | – | 55.1% | | 60.0% | 53.1% | 0.407 | 53.4% | 59.0% | 0.479 |
| Yes | 7.9% | – | 44.9% | | 40.0% | 46.9% | | 46.6% | 41.0% | |
| If you were ever unable to be with your newborn, was another parent, family members or caregiver able to be with your baby? | | | | | | | | | | |
| No | 14.2% | – | 80.9% | | 78.0% | 82.% | 0.539 | 87.1% | 68.9% | 0.003 |
| Yes | 12.6% | – | 19.1% | | 22.0% | 18.% | | 12.9% | 31.1% | |
| **Other care indicators** | | | | | | | | | | |
| Clinical quality index | | | | | | | | | | |
| Mean | 4.14 | 4.14 | 4.15 | 0.965 | 4.13 | 4.20 | 0.637 | 3.81 | 4.64 | <0.001 |
| SD | (1.37) | (1.35) | (1.45) | | (1.36) | (1.40) | | (1.38) | (1.17) | |
| Breastfeeding care index | | | | | | | | | | |
| Mean | 2.00 | 2.04 | 1.80 | 0.006 | 2.01 | 1.88 | 0.173 | 1.79 | 2.30 | <0.001 |
| SD | (1.07) | (1.05) | (1.13) | | (1.06) | (1.10) | | (1.10) | (0.94) | |

*Sample size, N=1013 (one refused to answer).
†For analyses, one missing response for newborn complications was recoded as 'healthy, no issues'.
GA, gestational age.

**Table 4** Distributions of outcomes by reports of newborn separation, separation for more than 10min and consent sought for newborn procedures (n=1014)

| Outcomes | Total | Newborn separated | | | Newborn separated >10min | | | Consent sought for newborn procedures* | | |
|---|---|---|---|---|---|---|---|---|---|---|
| | | Not separated | Separated | P value | Not separated or separated ≤10min | Separated >10min | P value | Did not ask for permission | Asked for permission | P value |
| Total number in group | 1014 | 836 | 178 | | 886 | 128 | | 607 | 406 | |
| Satisfaction with newborn care | | | | | | | | | | |
| Dissatisfied | 5.9% | 6.1% | 5.1% | 0.592 | 6.1% | 4.7% | 0.528 | 8.4% | 2.0% | <0.001 |
| Satisfied | 94.1% | 93.9% | 94.9% | | 93.9% | 95.3% | | 91.6% | 98.0% | |
| Attended postpartum visit after discharge, within 10weeks | | | | | | | | | | |
| No, did not attend | 53.1% | 53.5% | 51.1% | 0.569 | 53.4% | 50.8% | 0.581 | 57.7% | 46.1% | <0.001 |
| Yes, attended | 46.9% | 46.5% | 48.9% | | 46.6% | 49.2% | | 42.3% | 53.9% | |
| Exclusively breast feeding at follow-up 2–4 weeks† | | | | | | | | | | |
| No | 9.0% | 8.5% | 11.9% | 0.193 | 8.5% | 13.% | 0.128 | 9.8% | 8.0% | 0.373 |
| Yes | 91.0% | 91.5% | 88.1% | | 91.5% | 87.% | | 90.2% | 92.0% | |
| Exclusively breast feeding at 10-week follow-up‡ | | | | | | | | | | |
| No | 18.0% | 17.7% | 19.6% | 0.577 | 17.6% | 21.0% | 0.399 | 20.1% | 15.0% | 0.058 |
| Yes | 82.0% | 82.3% | 80.4% | | 82.4% | 79.0% | | 79.9% | 85.0% | |

*Sample size, N=1013 (one refused to answer).
†Sample size, N=829 (women who completed follow-up 2–4 weeks and reported on exclusive breast feeding).
‡Sample size, N=839 (women who completed 10-week follow-up and reported on exclusive breast feeding).

Table 5    Results of multivariate logistic regression analyses examining the associations between separation, consent for newborn care, and maternal and newborn outcomes (n=1014)

| Outcome | Newborn separated from mother | Newborn separated for more than 10 min | Consent sought for newborn procedures* |
|---|---|---|---|
| | aOR (95% CI) | aOR (95% CI) | aOR (95% CI) |
| Satisfaction with newborn care at baseline (*ref. not satisfied*) | 1.34 (0.75 to 2.41) | 1.36 (0.57 to 3.28) | 2.71 (1.67 to 4.41)*** |
| Attended a postpartum visit, within 10 weeks (*ref. no visit*) | 1.14 (0.96 to 1.35) | 1.11 (0.86 to 1.43) | 1.27 (1.05 to 1.55)* |
| Exclusively breast feeding at 2–4 weeks† (*ref. not exclusively breast fed*) | 0.67 (0.41 to 1.09) | 0.56 (0.40 to 0.76)** | 1.06 (0.78 to 1.44)‡ |
| Exclusively breast feeding at 10 weeks§ (*ref. not exclusively breast fed*) | 0.76 (0.49 to 1.18) | 0.66 (0.39 to 1.10) | 1.33 (1.10 to 1.62)** |

All estimates of ORs (and 95% CIs) were adjusted for age, multiparity, marital status, education, employment, women's birthplace, facility type, insurance status, health status and clinical quality of care index. Satisfaction with newborn care models also adjusted for maternal complications and newborn complications at baseline. Breastfeeding outcome models also adjusted for breastfeeding care index, newborn complications at baseline, and both newborn and maternal complications after discharge. Postpartum visit models also adjusted for maternal and newborn complications after discharge. Cluster robust SEs were used to account for clustering by facility.
"P<0.05, **p<0.01, ***p<0.001.
*Sample size, N=1013 (women who reported consent sought for newborn procedures, excluded one missing response).
†Sample size, N=829 (women who completed follow-up survey 2–4 weeks and reported exclusive breastfeeding status).
‡Sample size, N=828 (women who completed follow-up survey 2–4 weeks and reported exclusive breastfeeding status, excluded one missing response for consent sought for newborn procedures).
§Sample size, N=839 (women who completed 10-week follow-up survey and reported exclusive breastfeeding status).
aOR, adjusted OR.

who reported providers asked for permission before performing newborns' procedures or examinations were 2.7 times more likely to be satisfied with care (aOR=2.71, 95% CI: 1.67 to 4.41) and were also 27% more likely to attend a postpartum visit by either 2–4 weeks or 10 weeks (aOR=1.27, 95% CI: 1.05 to 1.55) after controlling for covariates. Report of consent for newborn procedures displayed no association with exclusive breast feeding at 2–4 weeks (aOR=1.06, 95% CI: 0.78 to 1.44), but was associated with a 33% increased likelihood of exclusive breast feeding at 10 weeks (aOR=1.33, 95% CI: 1.10 to 1.62) even after controlling for breastfeeding care index and other previously described covariates.

## DISCUSSION

The immediate postpartum period is a critical clinical period with high mortality risk and an important social period for maternal–newborn bonding. Whether positive or negative, most families will hold strong memories of experiences in this time, and it may impact their opinions about and willingness to seek care in the future, health practices in the home and long-term health outcomes. We sought to examine two newborn care experiences related to the experience of care, to understand the context of these experiences and to assess associations with relevant outcomes: care satisfaction, use and breast feeding. This study provides evidence that treatment of newborns is an important component of quality and is likely to influence newborn health and postpartum practices.

Importantly, this study found significant inequities related to treatment of women and their newborns. For example, being consented for newborn care was correlated with insurance coverage, indicating that poorer women and their newborns may not be treated equally by the health system with regard to their clinical care. Women with higher parity, born outside of Nairobi or Kiambu counties, and who experienced delivery complications were more likely to experience separation from their newborns. Mothers and newborns were most likely to be separated at government hospitals, where the most women in Kenya deliver. Although less common, separation did occur in the smaller health centre/dispensary as well as private facilities. Other studies in Kenya have found that while larger hospitals provide better clinical quality care than smaller hospitals, women are also more likely to experience poor patient-centred care.[50] Further research is needed to understand how much of separation is driven by facility policies rather than individual provider behaviour. Infrastructural issues like lack of space and insufficient infection control procedures may influence policies at health facilities to limit family member entry or separate newborns.[51] In underfunded facilities, fatigue and stress of health workers likely influence their ability to provide high-quality care.[52 53] On the other hand, providers may have been trained to promote practices that are no longer evidence based, such as misconceptions around separation for infection control.[54] Although drivers of poor quality and disrespectful care are not fully understood,[46] these findings

suggest that they are likely multifaceted and interactive. More research is needed in a wider range of settings and countries, to explore the potential contextual factors that might influence policies, behaviours and solutions.

In this study, 17.6% of women reported being separated from their newborns, which is less than that has been reported in other studies.[47] Even so, many of these occurrences were thought by women to be unnecessary, and almost 10% were not given a reason for separation. Most separations were over 10 min and in only one-fifth was another family member able to accompany the newborn. In many contexts, and Kenya, specifically, fathers have limited access to maternity and postnatal wards and neonatal intensive care units.[8] Notably, other studies have found that women in Nairobi were less likely to want a companion during labour and delivery compared with women in rural Kenya, Ghana and India; and were also less likely to report being allowed a companion all the time.[8 12] However, there is a potentially important role for fathers, family members or doulas, in staying with the newborn when mothers require rest, surgery or critical care.[27] Further research is needed to examine facility-level policies in allowing a companion of choice and facilitating family support when it is desired.

Alarmingly, over 60% of women reported not being consented for newborn procedures. Report of being asked to provide consent was correlated with higher scores on the clinical quality of care and breastfeeding indices. This may reflect that providers or facilities that are more consistent in the provision of high-quality care are also more likely to seek permission for procedures. This is consistent with another study in Kenya that found high levels of failure to provide informed consent and notable patient–provider discordance in reports of informed consent. Even in instances where providers reported obtaining consent from women for medical procedures, women were not provided full information or did not comprehend the information given to them.[14] Future research may also examine the extent to which consent practices were due to provider characteristics, such as position, level of training or age.

We found that for both satisfaction and postpartum care utilisation, report of being asked for permission for newborn care procedures was positively associated, further illustrating the importance of consent as an aspect of respectful care.

This study provides evidence that newborn treatment during childbirth is associated with newborn health and postpartum practices. Reported separation of newborns from their mothers for over 10 min was associated with decreased breast feeding at a later time point. While this association could be confounded by newborns with more severe complications—which require additional procedures, and also affect breast feeding—the association remained after adjustment in the model. Sensitivity analyses found no effect of low birth weight and preterm birth on estimates of association (and neither were statistically significant in models), suggesting that this association was not due to smaller or sicker infants who may have greater difficulty breast feeding. We also acknowledge the possibility that newborns may have been separated after the interview if they experienced later complications. While this study did not capture this information, we expect that this might have biased the estimated association to be more conservative.

Although the timing and duration of separation is likely important, we lacked information about the length of separation and when it occurred. Nevertheless, this finding suggests that any prolonged separation during the immediate postpartum period (within 48 hours) may inhibit critical bonding and breastfeeding processes. We highlight the need to investigate a critical threshold of separation and timing as an important area for future research.

It is also plausible that the lack of early initiation of breast feeding and lower satisfaction with immediate postnatal care contributed to poorer breastfeeding practices after discharge because of lower motivation or encouragement. In addition to improved satisfaction and likelihood to use postnatal care, report of consent for newborn procedures was associated with a greater likelihood of exclusive breast feeding at follow-up, indicating a potential causal link. In addition, given that women who did not report consent also reported lower clinical quality and breastfeeding care scores, these women likely experienced overall lower quality of care. These combined negative experiences of care may have also contributed to distrust of health professionals' advice to exclusively breast feed or greater biological stress responses that may impede breast feeding. This finding is important given the overwhelming evidence of the benefits of breastmilk for newborn survival, development and health of infants.[55 56] This adds to the evidence found by other studies which showed that lack of respectful and supportive care during intrapartum care is associated with more reports of newborn complications.[48]

### Strengths and limitations

This study is one of the first studies to include questions with the primary aim of understanding details around newborn separation, and to correlate newborn care experiences with subsequent health outcomes. While previous studies have reported on prevalence of newborn separation,[7 10 46] this study is the first to differentiate separation by maternal perception of reason and need, highlighting both the probability that much of the separation is not medically necessary, and that many parents do not fully understand the reasons for separation most likely due to poor communication. This study also goes beyond measuring maternal consent for care of self to consent of procedures for the newborn. Lastly, this study used longitudinal data and was able to measure treatment of newborns from predischarge care in facilities to postpartum practices.

We lacked information on the details of separation that may influence the magnitude of associations, including

the duration or timing of separation and information about what types of newborn procedures or examinations for which women reported lack of consent, such as whether they were routine, invasive, emergency procedures or involved admission to a newborn ward. This study was limited to 10 weeks of follow-up, and it is unknown how care practices or experience of care for newborns may influence care throughout infancy. Additionally, the samples for the follow-ups at 2–4 and 10 weeks differed slightly (though not significantly across sociodemographics or care), so our comparisons of exclusive breast feeding between these time periods may be interpreted conservatively due to smaller sample size and potential biases due to composition. Further, this study focused on women's reports of their care and did not triangulate with hospital records or observations. Thus, participants may have had recall bias or notable reporting error and results should be interpreted conservatively.[57] The negative framing of newborn experience questions may have also biased participants to respond more negatively to other questions, such as satisfaction. Participants may have defined 'separation' differently than the researchers, and may have different interpretations than healthcare providers about certain practices, such as necessity of separation. While it was explained to participants that this study was confidential and was being conducted independently of the facility or health system, there may have been desirability bias if respondents were concerned about their subsequent care at facilities. Lastly, our results may have limited generalisability, as our study sample included facilities from Nairobi and Kiambu counties, Kenya. Future research may explore whether associations with these care experiences are also present in other social and institutional contexts.

## CONCLUSIONS

Newborns, their mothers and their families have a right to high-quality, respectful care, regardless of their characteristics or where they are delivered. High-quality newborn care includes the ability to stay together and for the parents to be informed and consent for care. The implications of various practices on health outcomes a month or more after discharge illustrate the importance of a positive experience of postnatal care. More research and action are needed to provide newborns with the standard of care that they deserve.

**Acknowledgements** We would like to thank Innovations for Poverty Action–Kenya for their contributions to the conceptualisation and data collection. We thank Ginger Golub and Sun Cotter for assistance in collection of data. We also express gratitude and appreciation to all of the women who participated in this study.

**Contributors** MS and ES conceptualised and designed the study. MKN analysed the data. MKN, ES, JO and MS contributed to the writing and revision of the manuscript.

**Funding** This study was funded by the Bill and Melinda Gates Foundation (OPP1127467). The authors thank the California Centre for Population Research at UCLA (CCPR) for its training fellowship for Michelle Nakphong from the Eunice Kennedy Shriver National Institute of Child Health and Human Development (T32-HD007545).

**Competing interests** None declared.

**Patient consent for publication** Not required.

**Ethics approval** All study procedures were approved by the Institutional Review Boards at the University of California, San Francisco (protocol number 19-27783) and the Kenya Medical Research Institute (protocol KEMRI Non-SSC 666). Informed consent was obtained from all participants prior to participation.

**Provenance and peer review** Not commissioned; externally peer reviewed.

**Data availability statement** Data are available upon reasonable request. Data are available upon reasonable request. Please contact the corresponding author with requests for de-identified participant data.

**ORCID iD**
Michelle Kao Nakphong http://orcid.org/0000-0003-2632-8007

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
