## [Reviewer comments · BMJ Open]

ARTICLE DETAILS

TITLE (PROVISIONAL)	The Association between Newborn Separation, Maternal Consent, and Health Outcomes: Findings from a Longitudinal Survey in Kenya
AUTHORS	Nakphong, Michelle; Sacks, Emma; Opot, James; Sudhinaraset, May

VERSION 1 – REVIEW

REVIEWER	Madiba , Sphiwe Sefako Makgatho Health Sciences University, Department of Public Health
REVIEW RETURNED	27-Jan-2021

GENERAL COMMENTS	Thank you for the opportunity to review this interesting and enlightening manuscript. The manuscript examined separating newborns from mothers and maternal consent for newborn care. I commend the authors for this well written quantitative piece that highlight the correlation of newborn care practices with subsequent health outcomes. This has implications for maternal and child health programmes and interventions to reduce morbidity and mortality not only in Kenya, but in other settings in developing countries. I have only one comment for the authors, I found the title complex and not easy to understand.
--

REVIEWER	Burrowes, Sahai Touro University California
REVIEW RETURNED	28-Jan-2021

GENERAL COMMENTS	This manuscript was a pleasure to review. The findings deepen our understanding of the importance of respectful maternal care and addresses a notable gap in the literature. The study seems well-conceived and well-executed, and the manuscript was clearly written. I have no strong recommendations for revisions. I would have liked to have seen more information on the effect of /interactions between the covariates in the multivariable models to better understand the magnitude and significance of the different factors affecting the outcomes, but I understand that this might have made the manuscript unwieldy. Relatedly, I would have like to have seen a multivariate analysis of the maternal and newborn factors associated with separation and consent in addition to the bivariate tables but this might be better suited for another paper.
--

	As the authors note, it would be useful if future studies triangulated mothers' reports with observations and information on provider characteristics. For example, I do wonder if providers who seek consent from patients differ in meaningful ways than those who don't (e.g., in level of training, age, etc.)
--	---

REVIEWER	Day, Louise-Tina LSHTM
REVIEW RETURNED	15-Feb-2021

GENERAL COMMENTS	Thank you for inviting me to review this interesting article which generates important evidence about separation of newborns and their mothers and consent for procedures around the time of birth and association with health outcomes at longitudinal follow-up. Comments to the author:  1. Authorship Among 4 authors, one has a Kenya affiliation for an international NGO. Did you consider including Kenya colleagues as co-researchers/ co-designers of this important research question? I noticed that two other colleagues are acknowledged for assistance in data collection. 2. Women's report Your study is based on women's report of events around the time of birth, yet many studies have shown that accuracy of reporting around the time of birth is mixed (EN-BIRTH study being the latest to add to that evidence). Your study uses reported "exposures" and reported "outcomes" so suggest when you report the findings to remind the reader you are drawing your associations from "reported separation" and "reported health outcomes" etc. e.g. Discussion p21 line 26 "separation of newborns.....was associated with....". Suggest would be good to explore in more detail other reasons for these associations – linked with your data on reported newborn and maternal conditions (among others). This is mentioned in results page 14 line 17ff and could be expanded. 3. Time of baseline/ follow-up interview If I understand correctly the baseline interviews were on the ward or at exit within 48 hours of birth. Women may report "no separation" at the time of the interview and then experience separation from their infant? It wasn't clear to me which interview data were being presented – baseline or follow-up? 4. Separation You mentioned in your limitations (Discussion page 19 line 8) that participants may have defined separation differently than the researchers and I agree with this important point . As one of your two "exposures" it would help the reader to understand the details behind the question in Table 1. "Was your baby ever separated from you for more than 10 mins". Were any probing questions used? The question is phrased in a negative way – separation instead of togetherness. How might that have introduced bias? I am also curious what was the justification of using 10 minutes as the cut-off? Was this intended to be a literal 10 minutes or to represent a short/long period of separation? Is there any evidence that 10 minutes is a "significant" amount of separation in the literature (e.g comparing 9 minutes separation to 60 minutes separation?). In the introduction page 5-6 you describe mother-infant bonding and skin-to-skin care immediately after birth. I am curious why you
--

	asked the mother the question in this general way – does it matter when the separation occurs – 10 minutes in the first hour of life, might have a different effect than 10 minutes at any point during admission in hospital? Were these babies with their mother on a postnatal ward or admitted to a newborn ward? The finding of less exclusive breastfeeding (EBF) at 2-4 weeks with > 10 minutes breastfeeding is interesting – can you suggest why this might be the case? Is it possible that these were the smallest/ or sickest babies with other reasons to be not fully breastfed by then. Have you done analyses with the e.g. LBW groups? Please add definition of “necessary separation” to Table 1. 5. Consent for procedures As the other of your two “exposures” it would help the reader to understand the details behind the question in Table 1 and what the mother might have understood by “procedures or examinations” (Table 1) for the 59.9% who reported they were not asked for permission. In the abstract this is shortened to “procedures” which seems a medical term – are we talking about measuring temperature/ vital signs or something more invasive such as a blood test? You have the data for newborn complications and wonder if you have analysed “of those who reported not giving permission” do you know if these were admitted to a newborn ward? 6. Reasons for separation I agree with you that zero separation is ideal, however there are rare situations when a baby is critically ill why separation may occur. Results page 14 line 17 - Sentence beginning “more newborn complications were reported among those separated...” – could that be reverse causality – the complications were the reason for the separation? 7. Other ABSTRACT: please add how many health facilities were included in your study at which level (and also in methods in the manuscript INTRODUCTION: page 3, line 29 – current estimates are 2.4 million newborns estimated to die annually page 4, line 40 – consent for procedures or exams – suggest “parental consent” Page 4 line 49 – “verbally” – is crying a verbal expression of needs? Perhaps rephrase Page 5 line 54 – “likely to successfully breastfeed during their first feed” – suggest rephrase Page 6 line 3 – “breastfeeding effectiveness”? or effective breastfeeding or effective suckling? Page 6 line 22 – “caregivers” twice DISCUSSION: Page 19 line 8 – Two newborn care practices – suggest rephrase – I wasn’t clear if this meant outcome practices at first (e.g. EBF at 2-4 weeks) but think you mean the two exposures (separation and lack of consent) perhaps? In which case I think these aren’t “practices” in the way we call EBF or drying after birth a “practice” but rather a care experience or something similar? Page 22 line 35 – what do you mean by “later pediatric care”? Later care for the child perhaps?
--	--

VERSION 1 – AUTHOR RESPONSE

Reviewer: 1

Dr. Sphiwe Madiba , Sefako Makgatho Health Sciences University

Comments to the Author:

Thank you for the opportunity to review this interesting and enlightening manuscript. The manuscript examined separating newborns from mothers and maternal consent for newborn care. I commend the authors for this well written quantitative piece that highlight the correlation of newborn care practices with subsequent health outcomes. This has implications for maternal and child health programmes and interventions to reduce morbidity and mortality not only in Kenya, but in other settings in developing countries.

I have only one comment for the authors, I found the title complex and not easy to understand.

Response: Thank you for your interest in this manuscript and your comment regarding the title. We have edited the title for clarity: “The association between newborn separation, maternal consent, and health outcomes: findings from a longitudinal survey in Kenya.”

Reviewer: 2

Dr. Sahai Burrowes, Touro University California

Comments to the Author:

This manuscript was a pleasure to review. The findings deepen our understanding of the importance of respectful maternal care and addresses a notable gap in the literature. The study seems well-conceived and well-executed, and the manuscript was clearly written.

I have no strong recommendations for revisions. I would have liked to have seen more information on the effect of /interactions between the covariates in the multivariable models to better understand the magnitude and significance of the different factors affecting the outcomes, but I understand that this might have made the manuscript unwieldy. Relatedly, I would have like to have seen a multivariate analysis of the maternal and newborn factors associated with separation and consent in addition to the bivariate tables but this might be better suited for another paper.

Response: We have included full models of several multivariable models in supplement 3. We also agree that a multivariate analysis of the maternal and newborn factors associated with separation and consent would be useful but agree that it might be better suited for another manuscript.

As the authors note, it would be useful if future studies triangulated mothers’ reports with observations and information on provider characteristics. For example, I do wonder if providers who seek consent from patients differ in meaningful ways than those who don’t (e.g., in level of training, age, etc.)

Response: We agree that it would be useful to examine the extent to which consent practices and separation were due to provider characteristics. We acknowledge that we lacked information about provider characteristics. We note this in the Discussion and highlight it as an area of future research (p.21).

Reviewer: 3

Dr. Louise-Tina Day, LSHTM

Comments to the Author:

Thank you for inviting me to review this interesting article which generates important evidence about separation of newborns and their mothers and consent for procedures around the time of birth and association with health outcomes at longitudinal follow-up.

Response: Thank you for your thoughtful review and constructive comments.

Comments to the author:

1. Authorship

Among 4 authors, one has a Kenya affiliation for an international NGO. Did you consider including Kenya colleagues as co-researchers/ co-designers of this important research question? I noticed that two other colleagues are acknowledged for assistance in data collection.

Response: Our study team has worked with Kenyan colleagues for a number of years. The research organization which led all fieldwork (Innovations for Poverty Action - Kenya) is led by a Kenyan Country Director, and all researchers who participated in the conceptualization and data collection from IPA-Kenya were Kenyan except for one project manager (Ginger Golub who is listed in the Acknowledgements and is a Kenyan resident, but not Kenyan citizen). However, these colleagues did not meet the criteria for authorship of this manuscript. James Opot (co-author) is a Kenyan citizen and oversaw all data collection activities and contributed to the design of the study.

2. Women's report

Your study is based on women's report of events around the time of birth, yet many studies have shown that accuracy of reporting around the time of birth is mixed (EN-BIRTH study being the latest to add to that evidence). Your study uses reported "exposures" and reported "outcomes" so suggest when you report the findings to remind the reader you are drawing your associations from "reported separation" and "reported health outcomes" etc. e.g. Discussion p21 line 26 "separation of newborns.....was associated with....". Suggest would be good to explore in more detail other reasons for these associations – linked with your data on reported newborn and maternal conditions (among others). This is mentioned in results page 14 line 17ff and could be expanded.

Response: We have edited the language to reflect that both 'exposures' and 'outcomes' were based on women's reports. We also acknowledge that using only women's reports may have introduced reporting error and recall bias in the Discussion (p.24).

As the reviewer highlighted, women who reported not being consented for newborn procedures were more likely to report lower quality of care in other ways. It is possible that these combined experiences of poor quality of care led to greater mistrust of health professionals' advice to exclusively breastfeed or stress responses that could affect milk supply or impede breastfeeding. We added this point to our discussion (p. 22).

3. Time of baseline/ follow-up interview

If I understand correctly the baseline interviews were on the ward or at exit within 48 hours of birth. Women may report “no separation” at the time of the interview and then experience separation from their infant? It wasn’t clear to me which interview data were being presented – baseline or follow-up?

Response: It possible that separation happened after the interview if the baby developed complications after the interview but before discharge. This was not captured in this study, it is possible that we may not have this may have likely biased the estimated association between separation and exclusive breastfeeding at 2-4 weeks to be more conservative. We have acknowledged this limitation in the discussion (p.22).

We present data from both the baseline (collected after delivery) and follow-up interviews. Most data presented is from the baseline survey (e.g., newborn care experiences, sociodemographics, health status, satisfaction). The data from follow-up interviews include report of complications following discharge and outcome measures: postpartum visit and exclusive breastfeeding. We have added information about when measures were collected to Table 1.

4. Separation

You mentioned in your limitations (Discussion page 19 line 8) that participants may have defined separation differently than the researchers and I agree with this important point . As one of your two “exposures” it would help the reader to understand the details behind the question in Table 1. “Was your baby ever separated from you for more than 10 mins”. Were any probing questions used? The question is phrased in a negative way – separation instead of togetherness. How might that have introduced bias?

I am also curious what was the justification of using 10 minutes as the cut-off? Was this intended to be a literal 10 minutes or to represent a short/long period of separation? Is there any evidence that 10 minutes is a “significant” amount of separation in the literature (e.g comparing 9 minutes separation to 60 minutes separation?).

Response: Regarding separation, additional probing questions were not asked. Thank you for the question about the 10 minute “cutoff”. While it is unknown whether women measured their separation, it was intended to be a proxy for short and long separation. We did not find evidence in the literature about a “significant” amount of separation, but we agree this is an important area for future research and are hopeful this preliminary study can catalyse future work. We have added this information to the Methods (p.9) and Discussion (p.22).

We agree that the negative framing of the question may have introduced bias, especially if women tended to avoid cognitive dissonance and answer more negatively for other questions, such as satisfaction. We also note this in the Discussion (p.23)

In the introduction page 5-6 you describe mother-infant bonding and skin-to-skin care immediately after birth. I am curious why you asked the mother the question in this general way – does it matter when the separation occurs – 10 minutes in the first hour of life, might have a different effect than 10 minutes at any point during admission in hospital? Were these babies with their mother on a postnatal ward or admitted to a newborn ward? The finding of less exclusive breastfeeding (EBF) at 2-4 weeks with > 10 minutes breastfeeding is interesting – can you suggest why this might be the case? Is it possible that these were the smallest/ or sickest babies with other reasons to be not fully breastfed by then. Have you done analyses with the e.g. LBW groups? Please add definition of “necessary separation” to Table 1.

Response: While the timing of separation likely does matter given the critical period of bonding that occurs immediately after birth, the question was asked in a general way because women may not have an accurate sense of time, especially after delivery. We have noted how the lack of specificity in this question limits our ability to examine the possible timing effects in the Discussion (p. 22).

Normally, babies are usually with their mothers in the postnatal ward unless the baby develops a complication. In some cases, the infant is admitted to the newborn ward for examination or monitoring. Because we lacked information about admissions to a newborn ward, we do not know whether this type of separation influenced associations.

We considered confounding by low birth weight (<2.5kg), preterm birth (gestational age 37 weeks or less), and reported complications (at baseline or following discharge). Our analyses indicate that the association between separation and exclusive breastfeeding remained even after controlling for each of these factors. We were unable to conduct stratified analyses due to the small proportion of newborns who were preterm (22.7%) or low birthweight (2.8%). We have addressed this point in the Discussion (p.21-22).

We included “perceived necessary separation” and the corresponding question to Table 1.

5. Consent for procedures

As the other of your two “exposures” it would help the reader to understand the details behind the question in Table 1 and what the mother might have understood by “procedures or examinations” (Table 1) for the 59.9% who reported they were not asked for permission. In the abstract this is shortened to “procedures” which seems a medical term – are we talking about measuring temperature/ vital signs or something more invasive such as a blood test? You have the data for newborn complications and wonder if you have analysed “of those who reported not giving permission” do you know if these were admitted to a newborn ward?

Response: “Procedures or examinations” may have been understood broadly as encompassing any type of medical actions performed on the newborn for diagnostic or treatment purposes. We lacked information about whether participants were responding about routine, invasive, or emergency

procedures and examinations. We have added this point to the Discussion (p. 23), and note that this is another area ripe for future research.

We did not collect information about whether newborns were admitted to a newborn ward, and have noted this limitation in the Discussion (p.23).

6. Reasons for separation

I agree with you that zero separation is ideal, however there are rare situations when a baby is critically ill why separation may occur. Results page 14 line 17 - Sentence beginning “more newborn complications were reported among those separated...” – could that be reverse causality – the complications were the reason for the separation?

Response: We agree that newborn complications could have been the reason for separation in some cases. We rephrased this sentence to reflect this directionality (p.14).

7. Other

ABSTRACT:

please add how many health facilities were included in your study at which level (and also in methods in the manuscript

Response: Six facilities were included in the study. Three facilities were government hospitals, two were private hospitals, and one was a government health centre/dispensary. This information has been added to the abstract and methods section (p.8). This information has been added to the abstract and the methods.

INTRODUCTION:

page 3, line 29 – current estimates are 2.4 million newborns estimated to die annually

Response: Thank you for this correction. This estimate and citation have been updated (p.3).

page 4, line 40 – consent for procedures or exams – suggest “parental consent”

Response: We have clarified “parental consent.” (p.4)

Page 4 line 49 – “verbally” – is crying a verbal expression of needs? Perhaps rephrase

Response: We have edited this phrase to read, “use words to express their needs.” (p.4)

Page 5 line 54 – “likely to successfully breastfeed during their first feed” – suggest rephrase

Response: We have edited this to read, “adequately suckle during their first feed” (p.5)

Page 6 line 3 – “breastfeeding effectiveness”? or effective breastfeeding or effective suckling?

Response: We have edited these phrases to read, “score higher on suckling competence.” (p.6)

Page 6 line 22 – “caregivers” twice

Response: We changed the second “caregivers” to “they” (p.6)

DISUCSSION:

Page 19 line 8 – Two newborn care practices – suggest rephrase – I wasn’t clear if this meant outcome practices at first (e.g. EBF at 2-4 weeks) but think you mean the two exposures (separation and lack of consent) perhaps? In which case I think these aren’t “practices” in the way we call EBF or drying after birth a “practice” but rather a care experience or something similar?

Response: We edited this phrase to read “newborn care experiences” for clarity. (p. 19)

Page 22 line 35 – what do you mean by “later pediatric care”? Later care for the child perhaps?

Response: We have rephrased this sentence “care throughout infancy.” (p.23)

VERSION 2 – REVIEW

REVIEWER	Day, Louise-Tina LSHTM
REVIEW RETURNED	14-May-2021
GENERAL COMMENTS	Many thanks for revising the paper after peer-review. I am satisfied my comments have been addressed. Congratulations on this interesting study.